# Differentiating Ductal Adenocarcinoma of the Pancreas from Benign Conditions Using Routine Health Records: A Prospective Case-Control Study

**DOI:** 10.3390/cancers15010280

**Published:** 2022-12-31

**Authors:** Mohamed Zardab, Vickna Balarajah, Abhirup Banerjee, Konstantinos Stasinos, Amina Saad, Ahmet Imrali, Christine Hughes, Rhiannon Roberts, Ajith Vajrala, Claude Chelala, Hemant M. Kocher, Abu Z. M. Dayem Ullah

**Affiliations:** 1Centre for Tumour Biology, Barts Cancer Institute, Queen Mary University of London, London EC1M 6BQ, UK; 2Barts and the London HPB Centre, The Royal London Hospital, Barts Health NHS Trust, London E1 1BB, UK; 3Barts Pancreas Tissue Bank, Barts Cancer Institute, Queen Mary University of London, London EC1M 6BQ, UK; 4Centre for Cancer Biomarkers and Biotherapeutics, Barts Cancer Institute, Queen Mary University of London, London EC1M 6BQ, UK

**Keywords:** PDAC, symptoms, blood test, comorbidity, prediction model

## Abstract

**Simple Summary:**

Pancreatic ductal adenocarcinoma (PDAC) constitutes a devastating disease with late diagnosis and poor overall survival, complicated by clinical presentations similar to benign pancreatic diseases. We aimed to analyse clinical parameters for differentiating suspected PDAC from benign conditions. The study holistically explored the presenting symptoms and routine laboratory test results of pancreatic disease patients during their consultation at secondary/tertiary care, including their demographic, lifestyle and comorbidity characteristics. Developed on a prospectively recruited cohort, this is the first machine learning-based prediction model that differentiates PDAC patients from those with benign conditions with a high degree of recall (sensitivity) and precision (positive predictive value). The model could serve to support clinicians’ decisions when assessing patients with pancreatic pathology and separating potential malignant candidates from benign ones for urgent referral to a tertiary centre. This could improve upon the current UK guidelines enabling early detection of PDAC by developing a digital referral tool.

**Abstract:**

The study aimed to develop a prediction model for differentiating suspected PDAC from benign conditions. We used a prospective cohort of patients with pancreatic disease (*n* = 762) enrolled at the Barts Pancreas Tissue Bank (2008-2021) and performed a case-control study examining the association of PDAC (*n* = 340) with predictor variables including demographics, comorbidities, lifestyle factors, presenting symptoms and commonly performed blood tests. Age (over 55), weight loss in hypertensive patients, recent symptoms of jaundice, high serum bilirubin, low serum creatinine, high serum alkaline phosphatase, low red blood cell count and low serum sodium were identified as the most important features. These predictors were then used for training several machine-learning-based risk-prediction models on 75% of the cohort. Models were assessed on the remaining 25%. A logistic regression-based model had the best overall performance in the validation cohort (area-under-the-curve = 0.90; Spiegelhalter’s z = −1·82, *p* = 0.07). Setting a probability threshold of 0.15 guided by the maximum F2-score of 0.855, 96.8% sensitivity was reached in the full cohort, which could lead to earlier detection of 84.7% of the PDAC patients. The prediction model has the potential to be applied in primary, secondary and emergency care settings for the early distinction of suspected PDAC patients and expedited referral to specialist hepato-pancreatico-biliary services.

## 1. Introduction

Pancreatic ductal adenocarcinoma (PDAC) is an aggressive malignant disease of the pancreas with a dismal 5-year survival rate of 3–15% [1]. It continues to be a diagnostic and therapeutic challenge with little survival improvement over the decades. PDAC is associated with a poor prognosis due to several factors, including low incidence (~12 per 100,000), non-specific symptoms, late presentation, aggressive and resistant tumour biology with early distant metastasis, and a lack of specific and sensitive biomarkers or imaging of early disease [1,2]. The only possible curative option is surgical resection, with adjuvant therapy now offering better survival. [1]. Unfortunately, ~80% of patients present with unresectable disease [2], which makes PDAC a disease with drastically worse morbidity and mortality than benign pancreatic pathologies [3] and other pancreatic cancers [4].

Prospective and retrospective studies on PDAC have shown a low prevalence and predictive value of clinical symptoms [5], although jaundice and weight loss appear to have a strong association in clinical practice [6], as well as new-onset diabetes, which has been identified as a significant indicator of PDAC in the older population (>60 years) [1,4]. While serum carbohydrate antigen 19-9 (CA19-9) is widely used for monitoring PDAC treatment response and recurrence, its utility as a diagnostic biomarker for screening in the general population is limited due to low sensitivity and specificity [1]. Symptom-based pathways to diagnosis in primary care are usually complicated by comorbidities [7], and patients are delayed for referral to specialist consultation at secondary and tertiary care centres even when presenting with clinically associated symptoms such as back pain, diabetes and weight loss [8]. Even if the referrals are followed commensurate with national guidelines, the chances of detecting early, potentially curable disease is still minuscule since most patients in this clinical pathway present with metastatic disease [9].

Digital technologies enable future healthcare professionals to take advantage of automated referral pathways, suggested by algorithms that handle numerous variables from routinely collected clinical data and search for interactive combinations to predict target outcomes [10]. Indeed, several predictive algorithms for identifying PDAC risk groups have been developed from large primary care databases [7,11], mainly focusing on presenting symptoms and demographic characteristics. Such symptom-based cancer decision support tools (CDSTs) offer improved discriminatory ability; however, they are over-fitted for certain patient groups and require continuous refinement with the inclusion of additional features [7,11]. Other information which could enhance the performance, including pre-existing medical conditions and routine laboratory tests, are rarely considered in these algorithms. The data source can also be a factor as reported pre-diagnostic symptom spectrum appears to be different between secondary/tertiary care and primary care [7,8,11]. Patients with pancreatic diseases, malignant and non-malignant alike, demonstrate similar symptomatic presentation (e.g., jaundice) and biochemical profile (e.g., elevated CA19-9 or deranged Liver Function Tests) [1], therefore differentiating these two groups within CDST workflow is also an important but challenging issue to reduce false positive detection. As the treatment and prognosis of benign pancreatic diseases and pancreatic cancers are completely different, fast intervention is required to eliminate misdiagnosis, ensure correct diagnosis, and select treatment options [12].

In this study, towards the broader goal of developing a digital referral tool for suspected pancreatic cancer, we utilised medical histories from a prospective cohort of patients with various pancreatic diseases who were treated at a specialised hepato-pancreatico-biliary (HPB) clinic in the UK. As a first step to developing this tool, we focused on PDAC—the most dominant (~90% of cases) and lethal form of pancreatic malignancy. Other pancreatic cancers, such as pancreatic neuroendocrine tumours, are less common, have different presenting symptoms (e.g., hormonal secretion) and are generally associated with a better prognosis compared to PDAC [3]. We identified a compendium of discriminatory symptoms and commonly performed laboratory test results that could differentiate PDAC from benign pancreatic conditions. Then, we aimed to develop an optimal machine learning-based predictive algorithm that could guide referral decisions for suspected PDAC.

## 2. Materials and Methods

### 2.1. Study Setting

All data utilised for this research were collected from the Barts Pancreas Tissue Bank (BPTB; https://www.bartspancreastissuebank.org.uk, accessed on 21 September 2021) study [13], with written informed consent from patients recruited at Barts Health NHS Trust (BHNT). The BPTB is a repository of biospecimens and associated clinical data collected from patients (≥18 years) referred to the specialist clinic of HPB surgery at BHNT (Barts and the London HPB Centre) for confirmed or suspected malignant or benign diseases of the pancreas and other diseases of hepatobiliary origin. The biobank also recruits healthy volunteers to be used as a contemporaneous comparison cohort.

Data on BPTB participants’ ‘health’ features are entered into a bespoke, encrypted, secure database using a predefined 49-item questionnaire, of which 37 items are completed by trained tissue collection officers in a face-to-face interview during recruitment. These include demographic information, anthropometric measurements, persistent symptoms, comorbidities, regular use of specific medication, lifestyle behaviours, family life, and family history of pancreatic and other cancers. Another 12 items of information (with varying granularity) about provisional and confirmed clinical diagnoses, treatment including surgical and non-surgical interventions, and test results including blood tests, urine tests, imaging, and histopathology reports are populated from their electronic health records (EHR) data at BHNT. Data is further cross-checked and verified by trained clinicians to ensure accuracy.

### 2.2. Study Design

We carried out a case-control study to examine the association of PDAC diagnosis with a set of predictor variables in comparison to patients with non-malignant pancreatic diseases (Pancreatic non-Cancer; PnC) and subsequently developed a risk-prediction algorithm that may separate diagnosis of PDAC from PnC. We queried the BPTB database to extract relevant clinical data for all patients registered with the BPTB study between 1 January 2008 and 21 September 2021 and diagnosed with any pancreatic condition (*n* = 780 of 1371) by excluding healthy controls and those with non-pancreatic conditions (*n* = 588). A diagnosis of pancreatic neoplasm is recorded in the BPTB database using the ICD-O-3 (International Classification of Diseases—Oncology, 2013) multi-axial classification for the neoplasm site and histology; the non-neoplastic disorders of the pancreas are coded using the ICD-10 codes (2019) (Appendix A). The final diagnosis is recorded using hierarchical ascertainment mechanisms: histology reports, multidisciplinary team meetings outcomes, consultation notes from the oncological, surgical or gastroenterological specialist, findings from test results (e.g., radiological and endoscopic reports) and commissioning data sets diagnosis entry (ICD-10 codes prepared by BHNT clinical coders). The date on the first confirmatory histology report is used as the date of diagnosis. Otherwise, the earliest dates from the other evidence are used. Complex diagnoses or cases with incomplete data are agreed on by an adjudication group of clinicians (HMK, KS, MZ).

Patients were finally assigned to one of the three groups (in the order of priority)—PDAC, other pancreatic cancers (PC), and PnC, based on their confirmed final diagnosis entry in the BPTB database. For each individual within a specific group, the date associated with the earliest diagnosis was considered the index date. This is particularly applicable for PnC group patients with multiple diagnoses of pancreatic diseases. Patients were excluded when they had a secondary tumour to the pancreas (*n* = 2) or concomitant primary cancer in another body site (*n* = 6) or were assigned a provisional diagnosis without a subsequently confirmed diagnosis (*n* = 1) to avoid biases (Figure 1). Comorbidity and symptoms history could not be collected for nine patients. Pancreatic cancer patients without a diagnosis of PDAC were further excluded (*n* = 58), in line with the study focus.

### 2.3. Assessment of Predictor Variables

Demographic details included in the study were gender (Male, Female); age (<55, 55–64, 65–74, 75+); ethnic origin (Caucasian, Afro-Caribbean, South Asian, or Other). Lifestyle variables included smoking history or alcohol consumption (current, past, or never) and body mass index (BMI). Clinical variables relating to comorbidities (present or absent) comprised diabetes, hypertension, high cholesterol, chronic respiratory disease, cardiovascular disease, chronic kidney disease, liver disease and previous cancer diagnosis. Both lifestyle and comorbidity variables were recorded based on the patient’s status at the time of recruitment in the BPTB.

The following eleven symptoms that might herald a diagnosis of gastrointestinal problems were included as binary status (present vs absent): pain, jaundice, weight loss, nausea, vomiting, diarrhoea, constipation, fatigue, loss of appetite, pruritus, and steatorrhea. The presence or absence of these symptoms within the one-year period before recruitment was confirmed in a face-to-face interview with patients. A composite symptom was also derived in the form of a change in bowel habits, manifested in any diarrhoea, constipation, or steatorrhea.

Twenty-two laboratory tests, commonly requested by clinicians for suspected hepatic-pancreatic-biliary (HPB) problems, were examined (Appendix A). For each test, the instance closest to the recruitment date and conducted within six months before the recruitment was considered. Otherwise, the first instance within thirty days of recruitment was used.

### 2.4. Statistical Analyses

Differences in demographic and clinical characteristics between the PDAC and PnC groups were assessed using Pearson’s Chi-square test or Kruskal–Wallis rank sum test, as appropriate. Predictor variables with data missing in more than 25% of the final study population (*n* = 704) were excluded from further analyses. We conducted the preliminary observational association study and the subsequent development of the risk-prediction algorithms on 75% of the data (derivation set). We then assessed the prediction algorithms on the remaining 25% (validation set). Both derivation and validation sets had similar proportions of PDAC and PnC patients. Multiple imputations by the method of the chained equation were used to replace missing values for predictor variables [14]. Five imputations were carried out separately in derivation and validation datasets. In the imputed datasets, results from each blood test were stratified by the known normal range of respective pathology test guidelines (Appendix A) and included in the analyses as categorical variables with three possible categories: low, normal and high. PDAC risk associated with individual predictor variables was evaluated with odds ratios (OR) with 95% confidence intervals (CI), using separate multivariable logistic regression models, and controlled for potential confounding effects by demographic variables—age group, gender, and ethnicity. Rubin’s rules were used to combine the results across the imputed datasets [15]. Considering that the reference group consists of pancreatic patients rather than the otherwise general population, we conducted additional risk assessments for individual blood tests: (i) stratified by the interquartile range (IQR) of known normal range [low, IQR-normal, high] (Appendix A), capturing additional blood tests with a potentially distinctive profile in the lower or upper range of normal values; (ii) as a continuous variable, capturing those tests with a similar profile in stratified ranges but distinctive clustering within those ranges.

Several risk-prediction models were developed utilising the findings from observational association investigation. All predictor variables having at least one category with high statistical significance (*p* < 0.01) after correction for multiple testing were retained to be used in the development of prediction models. Finally, with an aim to capture the effect of presenting symptoms and common blood tests in specific patient subgroups, we examined their interactions to encompass all demographic, comorbidity and lifestyle variables; all statistically significant two-way interactions (*p* < 0.01) were subsequently included. Multivariate logistic regression models were fitted within the supervised machine-learning setting. The outcome was the probability that a patient would develop PDAC, derived as a function of the predictor variables. Five variations of logistic regression (LR) models were fitted with: no penalty function, lasso regression with L1 regularisation penalty (LR-Lasso), ridge regression with L2 regularisation penalty utilising Bayesian Information Criterion (LR-Ridge), elastic net regression with L1/L2 regularisation penalty (LR-ElasticNet), and stepwise model selection by backward elimination utilising Akaike Information Criterion (LR-StepAIC). The average receiver operating characteristic (ROC) statistic from repeated 10-fold cross-validation training runs was used as the metric to determine the optimal hyperparameters for the prediction models. The regression coefficients in the final LR, LR-Lasso, LR-Ridge and LR-ElasticNet models were pooled from preliminary models fitted to the imputed datasets. The final LR-StepAIC model was a regular LR model fitted with a reduced set of predictor variables after applying a voting-based variable selection technique to the five LR-StepAIC models on the imputed datasets—variables were excluded if they did not appear in all the preliminary models.

The models’ performance were assessed based on the pooled outcome probability from five multiply imputed validation datasets. The best model was chosen based on the triple criteria of discrimination (area under the receiver operating characteristic curve [AUROC] statistic), calibration (Spiegelhalter’s Z-test) and accuracy (*F*_2_-score). *F*_1_-score is the harmonic mean of precision (positive predictive value) and recall (sensitivity) with equal importance. *F*_2_-score attaches twice as much importance to recall as precision, i.e., reducing false negatives at the cost of an increased false positive detection, which is an important consideration for this study while distinguishing PDAC patients from non-malignant pancreatic condition patients rather than “otherwise healthy” control population. Once the best model was selected, it was applied to the full dataset to extract the *F*-scores and decide on the optimal probability threshold for identifying potential high-risk PDAC cases. Further subgroup analyses were conducted to test the performance of the resultant model in differentiating PDAC from PnC subgroups: acute conditions, chronic conditions, benign tumours and cysts (Appendix A).

All statistical analyses and visualisations were performed in R (version 3.5.1).

## 3. Results

### 3.1. Descriptive Analyses

Overall, the study included 344 patients with a primary diagnosis of PDAC and 360 patients with PnC (Figure 1). The PnC group consisted of patients diagnosed with chronic pancreatitis (*n* = 113, 31%), acute pancreatitis (*n* = 90, 25%), benign tumour (*n* = 86, 24%), pancreatic cyst and pseudocyst (*n* = 57, 16%); and other benign diseases (Appendix A). Appendix A presents the baseline characteristics of the two groups. PDAC patients were significantly older compared to PnC (median 68 vs 55 years; *p* < 0.001) and had more prevalent comorbidities, particularly diabetes (*p* = 0.002) and high blood pressure (*p* < 0.001). Both groups had similar majority representation from White patients, but South Asian representation was notably lower in the PDAC group (4.9% vs 10.3%; *p* < 0.001). At the time of recruitment (closer to the time of index diagnosis), PDAC patients had relatively lower BMI compared to PnC patients (*p* = 0.002). Histories of smoking and alcohol consumption were marginally less in PDAC than in the PnC group, with lower current smokers (18.6% vs 27.2%) and past drinkers (12.8% vs 21.1%) among PDAC patients.

In the year prior to recruitment, jaundice and weight loss were reported more frequently by PDAC cases, while pain, nausea, vomiting and diarrhoea were more common among PnC patients (Appendix A). The biochemical profiles of PDAC patients show a significant association with higher levels of ALT, MCV, PLT, NEUT, and CRP (all *p* < 0.01) and lower levels of ALB, AMY, CREA, SOD, RBC, HB and LYMP (all *p* < 0.01; Appendix A). PDAC patients demonstrated nearly two times higher levels of ALP (median 162.5 vs 86) and TBIL (median 14.5 vs 8.0) than PnC patients. Unsurprisingly, being a known pancreatic cancer biomarker, CA19-9 levels were very high among PDAC patients compared to PnC patients (median 294.0 vs 15.9; *p* < 0.001).

### 3.2. Observational Association Study

The derivation dataset was used to inspect the association between predictor variables and odds of PDAC (*n* = 258) in comparison to PnC (*n* = 270). The following predictor variables were excluded from the analyses due to missing data in more than 25% of patients: AMY (77.6%), AST (62.4%), CA19-9 (46.6%), CA (40.1%), and CRP (31.4%). After controlling for demographic variables in the regression models, no comorbidity or lifestyle variables were independently associated with PDAC compared to PnC. Increased odds were observed for the increasing age group (OR 3.6–10.1; *p* < 0.001), jaundice and weight loss (~3-fold; *p* < 0.001) (Figure 2). Compared to the normal levels of blood test results, increased odds of PDAC diagnosis were significantly associated with elevated levels of ALP (OR 3.9 [2.6–6.1]; *p* < 0.001), ALT (OR 2.6 [1.5–4.4]; *p* = 0.002), and TBIL (OR 4.8 [2.9–8.1]; *p* < 0.001), and reduced levels of LYMP (OR 1.9 [1.2–3.0]; *p* = 0.008), CREA (OR 3.1 [1.4–6.6]; *p* = 0.008) and NA (OR 4.2 [1.6–10.6]; *p* = 0.006) (Figure 3a). In addition, we observed increased odds of PDAC associated with higher PLT levels with reference to IQR-normal level (OR 2.4 [1.3-4.3]; *p* = 0.007) (Figure 3b); conversely, each unit increase in RBC was associated with lower odds of PDAC (OR 0.44 [0.30–0.63]; *p* < 0.001) (Figure 3c). Unadjusted odds ratios are reported in Appendix A. There was also evidence to suggest that PDAC risk associated with several predictors may vary with lifestyle or comorbidity status, particularly recent weight loss appearing to be a significant indicator of PDAC in hypertensive patients (OR 3.8 [1.6–8.9]; *p* = 0.005) (Appendix A). However, gender, ethnic or age differences did not appear to modify the effect of any clinical parameters in differentiating PDAC from non-malignant cases.

### 3.3. Prediction Models for Differential Diagnosis of PDAC

Statistically significant predictor variables and interactions from the observational association study were included to develop the prediction models: age group, jaundice, weight loss, ALP, ALT, CREA, RBC, LYMP, PLT, NA, TBIL, and hypertension—weight loss interaction. RBC was a continuous variable; PLT was a categorical variable with IQR-normal as a reference; all other blood tests were categorical variables with a known normal range as a reference.

The multiply imputed validation datasets comprised 86 unique PDAC cases and 90 PnC patients. The AUROC curves of all prediction models were identical and significantly discriminatory, ranging between 0.89 and 0.90. (Table 1). After combining with calibration statistic and *F*_2_-score, the LR-StepAIC model stood out as the best-performing model (AUROC = 0.90; Spiegelhalter’s z = −1.8, *p* = 0.07; maximum *F*_2_-score = 0.89) with a smaller predictor set (Table 2). Hence, the LR-StepAIC model was selected as our final model, indicating good differentiation between PDAC and PnC cases whilst having close correspondence between predicted PDAC risk and observed outcome.

The maximum *F*_2_-score from the LR model applied to the full dataset was obtained for probability threshold at 0.15, in which case 81.1% (*n* = 571/704) of the study population would undergo urgent referral with a PDAC sensitivity of 96.8%. With a biomarker panel of 87.5% sensitivity [16] applied, this could lead to early detection of around 84.7% (*n* = 291/344) of PDAC patients (Figure 4). For a higher probability threshold of 0.42 from the maximum *F*_1_-score, 52.4% (*n* = 369) of the population would undergo urgent referral with an early detection rate of 71%. Table 3 presents the sensitivity, specificity and predictive values for different probability cut-offs obtained from maximum *F*_1_ and *F*_2_ scores and comparison with current National Institute for Health and Care Excellence (NICE) referral guidelines for suspected pancreatic cancer through specialist appointment within two weeks or urgent imaging within two weeks [17]. Applying the same probability threshold to test the model performance in differentiating PDAC particular PnC sub-groups revealed the same sensitivity (0.81) but a worsening specificity starting from the highest when differentiating with acute pancreatic (0.79) diseases to the worst when differentiating with benign tumours (0.70). (Appendix A).

## 4. Discussion

We conducted a case-control study on a patient cohort prospectively recruited at a tertiary HPB centre following a referral from primary care. We identified features in pancreatic patients’ recent medical history that could separate putative pancreatic ductal adenocarcinoma cases from benign pancreatic conditions, which present similarly clinically and are included as differential diagnoses. We utilised the findings to develop and validate a new prediction algorithm for differential diagnosis. To our knowledge, this is the first prospectively recruited study holistically exploring the presenting symptoms and common laboratory test results of pancreatic disease patients, at the time point of their consultation at secondary/tertiary care, within the context of their demographic, lifestyle and comorbidity characteristics. The derivation dataset is well proportioned with a large number of PDAC patients comparable to other prospective studies [8,18]. The final algorithm showed good discrimination and calibration on a separate validation dataset.

The study reaffirms previously known associations with PDAC, such as age, symptoms of jaundice and weight loss [5,11], as well as raised serum bilirubin (various degrees of jaundice), deranged liver function tests (blockage in intra- and extra-hepatic ducts), elevated platelet counts (hypercoagulable state), decreased lymphocytes (immune suppression), and decreased RBC (nutrient deficiency and anaemia) [1,4,19]. While the platelet profile appeared to be similar in PDAC and PnC patients, adjusting the reference range revealed PDAC association with elevated platelet levels, suggestive of many PDAC patients’ platelet levels being found in the higher end of the normal range. Similarly, switching to real values of blood test results confirmed that RBC can also be used as a parameter for differential diagnosis of PDAC and PnC, considering pancreatic patients generally show lower levels of RBC. This opens the door to further investigations regarding the link between lower RBC counts and PDAC, which has not been explored in the literature.

We also found novel associations with PDAC when compared to PnC, which will likely translate to amplified associations when compared to healthy individuals. Low sodium, which has previously not been reported in the PDAC literature, may reflect undiagnosed SIADH (syndrome of inappropriate antidiuretic hormone) in PDAC patients or perhaps due to cancer pathophysiology [20]. This will need further investigation into clinical correlation and may allow understanding of the disease process better. Raised ALP in PDAC patients may reflect biliary tree injury secondary to biliary obstruction. It is not classically a blood test used to differentiate between PDAC and benign diseases. Another novel finding was the association of low creatinine with PDAC, which may be due to liver damage through obstructive jaundice or disease-associated sarcopenia and cachexia leading to reduced muscle mass [21]. However, we found no association between higher BMI and PDAC, unlike what the literature suggests [3,8,11]. This could be due to the use of PnC patients as a comparison cohort, suggesting that obesity is likely to be associated with pancreatic conditions in general rather than disease severity [22]. We used patients’ most recent pre-diagnostic BMI rather than historical trends, which could mask PDAC patients’ transition from higher to lower BMI status. Similar findings suggest that diabetes alone cannot be considered as any better indicator for the manifestation of PDAC than other non-malignant conditions such as pancreatitis [23]. While any synergistic effect involving patient demographics could be important from the screening/surveillance perspective, we could not find any clinical parameter predictive of PDAC in certain gender or ethnic groups, suggesting that the severity of pancreatic conditions is dependent on other risk factors rather than gender or ethnic differences.

Our final prediction algorithm shows that age (over 55), weight loss in hypertensive patients, recent symptoms of jaundice, high serum bilirubin, high serum ALP, low RBC count and low serum sodium are the most important features when differentiating putative PDAC cases from less severe pancreatic conditions. The power of machine learning techniques allowed us to identify multiple weak indicators, which became only predictive when used in complex combinations with each other. We showed that using a modest separation probability threshold of 0.42 derived from the maximum *F*_1_-score could lead to the early detection of around 71% of the PDAC cases within our study cohort. An 81% sensitivity is still better than known diagnostic biomarkers (median sensitivity of 79%) [24]. This would optimise the use of the urgent referral pathway and/or costly investigative procedures by maximising the identification of potential PDAC cases at the cost of slightly increased false positive detection of non-cancerous pancreatic patients. Interestingly when dividing the PnC cohort into sub-groups based on the pathology, we see a good performance versus all groups indicating the viability of the model for differentiating from all benign diseases. We know that benign tumours have the most in common with PDAC pathophysiologically in these sub-groups, with chronic pancreatitis next as both a predominant risk factor and a similar disease. Pancreatic cysts and pseudocysts are sequelae to acute and chronic pancreatitis, and these results highlight the fidelity of the discrimination in the model. Considering the well-documented probability of progression to PDAC from chronic pancreatitis and benign pre-malignant tumours [1,2], early detection of these patient groups and taking appropriate measures can be considered as a cancer preventive strategy serving a broader goal.

It is important to note that prediction algorithms are often developed to identify pancreatic cancer in the general population with a significant representation of healthy controls [3,8,11,19,25], whereas our comparison cohort consisted of benign pancreatic disease patients. Hence, a direct performance comparison may not be appropriate, yet it is comparable to regression-based established prediction algorithms such as QCancer Pancreas (AUROC: 84–92% vs 86.2%) [25] or even ensemble-learning-based complex algorithm (*F*_1_-score: 64% vs 78%) [19]. The established protocols, such as the current NICE guidelines, show a very low sensitivity of 35–49% had these been applied to our study population. This possibly explains why a majority of pancreatic cancer cases are still diagnosed as a result of an emergency hospital presentation [11].

Key strengths of our study include the study design, with adherence to the STROBE guidelines and TRIPOD statement, and a prospectively recruited cohort with rich annotated clinical information from a multi-modal data collection process. While retrospective accounts of patient-reported medical history, in particular presenting symptoms, generally induce information bias [26], this had been greatly minimised by the sustained effort of trained clinicians at the biobank who went through unstructured clinical notes, pre-operative assessments, discharge summaries, or scanned documents for data triangulation. Another strength is the healthy representation of patients of non-White origin (21%), thereby increasing the generalisability of the results. This is the only study in our knowledge that has compared PDAC with other pancreatic disease at the time of presentation. As the recruited cohort of patients had consented to tissue and blood samples donation to the biobank, further biological studies can be performed, and a better prediction model can be derived that includes novel biomarkers, tissue histology, imaging results, genomic profiling, and survival and operative findings.

A key limitation of the study is the amount of missing data for blood test variables. We used a broader time window to collect participants’ blood test results, yet not all patients who visited the specialist HPB centre with suspected pancreatic conditions had those blood tests performed within the window. We attempted to minimise the impact by removing variables with >25% missing data and conducting analyses on multiple sets of imputed data. Yet, the final algorithm’s performance could improve further with the inclusion of CA19-9, carcinoembryonic antigen and amylase results. The reported blood test results were also not adjusted for ongoing medical interventions that could have affected our results, although most PDAC patients were recruited before diagnosis and intervention. Another limitation is the risk of unmeasured residual confounding. For example, the confounding effect of diabetes on PDAC risk could be different if participants with new-onset diabetes could be separated from those with long-standing diabetes. The overall study population was smaller in comparison to other retrospective cohort studies in the general population. However, with the objective of separating a low-prevalence difficult-to-diagnose malignant disease from its non-malignant counterpart, we had a balanced representation of cases and controls with reliable and accurate data. In order to gain sufficient power to take full advantage of machine learning, we acknowledge the need for a much larger prospective study. This could be logistically challenging to achieve. However, biobanks such as the Barts Pancreas Tissue Bank can support the goal with continuous recruitment of patients following a standardised and ethically approved protocol.

## 5. Conclusions

Risk prediction algorithms are an important tool in healthcare and help patients with time-sensitive diseases with early diagnosis and treatment, improved survival and quality of life. By incorporating widely available and easy-to-request blood tests, symptoms that patients should recognise when they occur, well-recorded comorbidities, and demographic details, our prediction algorithm may become a useful adjunct for primary care physicians to decide on the appropriate use of urgent referral or imaging pathway for suspected pancreatic patients. As half of the pancreatic cancer patients are still diagnosed through the emergency presentation route, an algorithm such as ours may also help appropriately triage patients at secondary care for expedited diagnosis and treatment. This has also created the groundwork to develop an easy-to-understand, utilisable risk score since it was trained through logistic regression rather than neural networks or ensemble learning models. However, we cannot ignore the need for further rigorous external validation before clinical implementation.

## Figures and Tables

**Figure 1 cancers-15-00280-f001:**
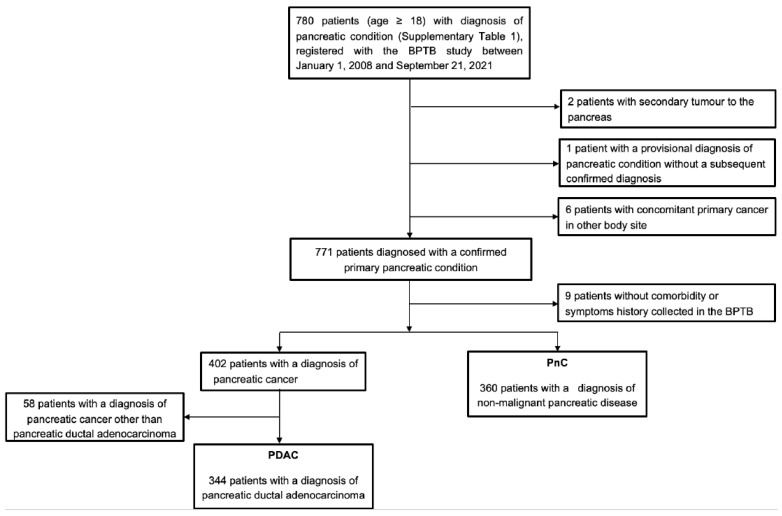
CONSORT diagram for selection of patients in the case-control study.

**Figure 2 cancers-15-00280-f002:**
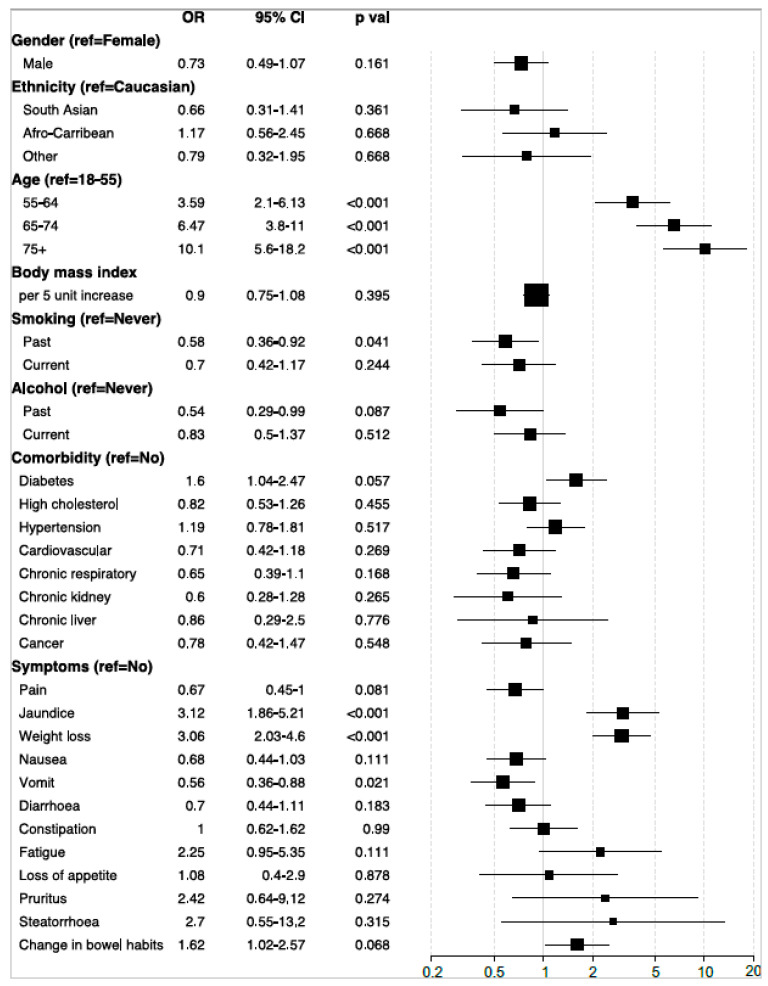
Forest plot showing the association between study variables (demographic, lifestyle, comorbidity and symptom) and odds of PDAC in comparison to the non-malignant pancreatic disease group. The odds ratio (OR) and 95% confidence interval (CI) are derived from the logistic regression model controlled for gender, ethnicity and age group. The reported p values are corrected for multiple testing via the Benjamini–Hochberg method.

**Figure 3 cancers-15-00280-f003:**
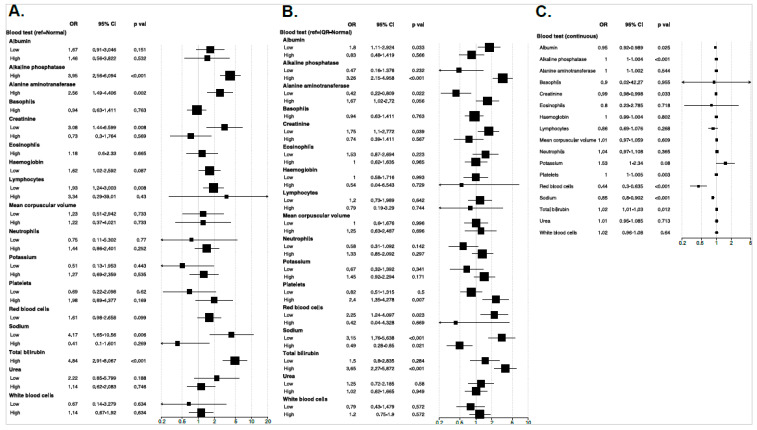
Forest plot showing the association between blood test result variables (**A**): categorical with the known normal range as a reference; (**B**): categorical with interquartile of known normal range as a reference; and (**C**): continuous and odds of PDAC in comparison to non-malignant pancreatic disease group. The odds ratio (OR) and 95% confidence interval (CI) are derived from the logistic regression model controlled for gender, ethnicity and age group. The reported p values are corrected for multiple testing via Benjamini–Hochberg method.

**Figure 4 cancers-15-00280-f004:**
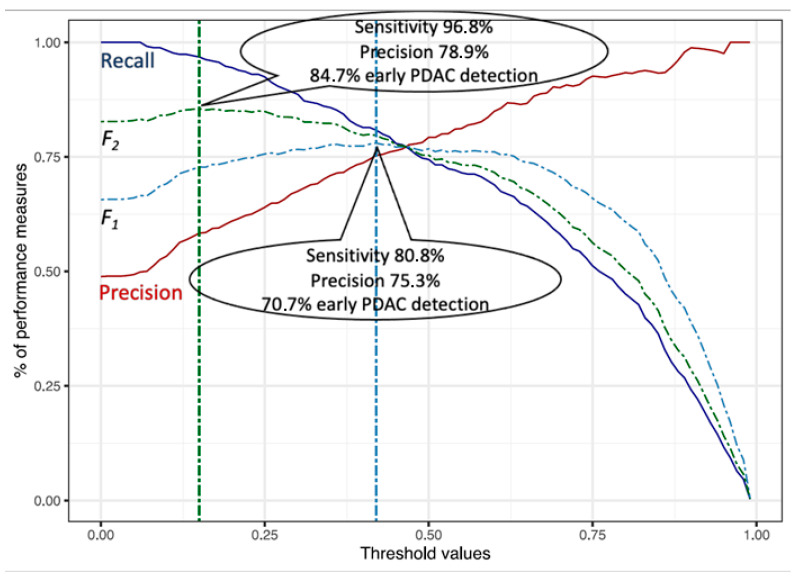
A performance measure graph of the LR-StepAIC model, with the resultant performance of the model at maximum *F*_1_ and *F*_2_-scores on the full dataset. The percentage of early PDAC detection includes the use of an 87.5% sensitive biomarker.

**Table 1 cancers-15-00280-t001:** Comparison of AUROC (area under the receiver operating characteristic) curve scores, Spiegelhalter’s z-test statistics and maximum *F*_2_-score for all models on the validation dataset (*n* = 176).

Model	AUROC		Spiegelhalter’s z Test		Maximum *F*_2_-Score
	Area	95% CI	Statistic	*p* Value	
LR	0.89	0.84–0.94	**−1.51**	**0.13**	0.89
LR-StepAIC	**0.90**	**0.85–0.94**	−1.82	0.07	**0.90**
LR-Ridge	0.89	0.85–0.94	−2.06	0.04	0.88
LR-Lasso	0.89	0.84–0.94	−1.60	0.11	0.89
LR-ElasticNet	0.89	0.85–0.94	−2.28	0.02	0.89

LR: logistic regression with no penalty function; LR-StepAIC: stepwise logistic regression with backward elimination utilising Akaike Information Criterion; LR-Ridge: ridge logistic regression with L2 regularisation penalty utilising Bayesian Information Criterion; LR-Lasso: lasso logistic regression with L1 regularisation penalty; LR-ElasticNet: elastic net logistic regression with L1/L2 regularisation penalty.

**Table 2 cancers-15-00280-t002:** Composition of the final PDAC risk prediction algorithm based on logistic regression with stepwise model selection by backward elimination utilising Akaike Information Criterion (LR-StepAIC).

	β Coefficient	Standard Error (SE)	Odds Ratio (OR)	95% Confidence Interval (CI)
**Intercept**	0.40	0.89	1.49	0.26–8.49
**Demographics**				
Age group (ref = <55)				
55–64	1.02	0.32	2.77	1.49–5.15
65–74	1.77	0.32	5.86	3.12–11.03
75+	2.04	0.36	7.66	3.78–15.55
**Symptoms (ref = No)**				
Jaundiced	0.67	0.27	1.95	1.14–3.34
Weight loss	0.35	0.29	1.42	0.8–2.52
**Blood tests** (ref = Normal)				
Alkaline phosphatase (ALP)				
High	0.70	0.25	2.01	1.23–3.28
Creatinine (CREA)				
Low	0.66	0.43	1.94	0.83–4.51
High	−0.88	0.50	0.41	0.16–1.1
Red blood cell count (RBC) ^a^	−0.60	0.19	0.55	0.38–0.79
Sodium				
Low	1.14	0.51	3.12	1.16–8.43
High	−0.22	0.64	0.8	0.23–2.79
Total bilirubin				
High	0.82	0.30	2.27	1.25–4.13
**Interactions** ^b^				
Hypertension				
without weight loss	−0.28	0.31	0.76	0.41–1.39
with weight loss	1.06	0.36	2.9	1.42–5.92

^a^ RBC is considered as a continuous variable with values to be used as a multiple of 10 ^12^ /L. ^b^ patients with no hypertension without weight loss are used as reference.

**Table 3 cancers-15-00280-t003:** Performance measures of the final model with different evaluation criteria (maximum *F*_1_ and *F*_2_-score) and comparison with NICE referral guidelines for suspected pancreatic cancer (*n* = 704).

	LR-StepAIC		Two-Week Wait ^a^	Urgent Imaging ^b^
	*F*_1_ Criteria	*F*_2_ Criteria		
Maximum *F*-score	0.780	0.855	0.603	0.483–0.519
Probability threshold	0.420	0.150	-	-
Sensitivity/Recall	0.808	0.968	0.488	0.352–0.390
Specificity	0.747	0.339	0.875	0.894–0.900
Precision/positive predictive value	0.753	0.583	0.789	0.771-0.779
Negative predictive value	0.803	0.917	0.642	0.592–0.605
Patients identified for referral	369	571	193	157–172
Proportion identified for referral	0.524	0.811	0.274	0.223–0.244
Maximum proportion of early detection of cancer ^c^	0.707	0.847	0.427	0.308–0.341

^a^ According to NICE guidelines for a specialist appointment within two weeks for patients aged 40 or over with jaundice. ^b^ According to NICE guidelines for urgent direct access CT scan (to be performed within 2 weeks), or an urgent ultrasound scan if CT is not available, for patients aged 60 or over with weight loss and any of the following: diarrhoea, pain, nausea, vomiting, constipation, new-onset diabetes. As a distinction cannot be made between diabetes and new-onset diabetes from our data, results are represented as a range derived from two sets of calculation: (i) excluding new-onset diabetes from the formula; (ii) diabetes variable as a placeholder for new-onset diabetes. ^c^ Considers a biomarker panel of 87.5% sensitivity.

## Data Availability

The Barts Pancreas Tissue Bank (BPTB) has full access to all the participants’ data in the study. Data used in this study are available via BPTB on request. A minimal dataset for replicating the reported findings is available via a request to corresponding authors.

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
