# Peer review of "Differentiating Ductal Adenocarcinoma of the Pancreas from Benign Conditions Using Routine Health Records: A Prospective Case-Control Study"

_cancers, 2022, doi:10.3390/cancers15010280_

Round 1
Reviewer 1 Report
In the current study titled “Differentiating ductal adenocarcinoma of the pancreas from benign conditions using routine health records: A prospective case-control study”, the authors developed a machine learning based prediction model that differentiates PDAC patients from those with benign conditions with a high degree of recall and precision based on a prospectively recruited cohort. This model has the potential to support clinicians’ decisions when assessing patients with pancreatic pathology and separating potential malignant candidates from benign ones in emergency situations.
The major problem with PDAC is late diagnosis and poor prognosis. This results in very poor 5-year survival rate of the PDAC patients. This is also impacted by the confusion between symptoms of patients with pancreatic malignacies and other non-malignant pancreatic pathological conditions. This prediction tool has the potential to overcome this and helps clinicians to take the further directions quickly in terms of treatment.
I congratulate the authors for their work.
No revision is required.
Reviewer 2 Report
Method for early diagnosis is the major challenge in the treatment of PDAC. Thus, the authors have identified a very important topic in the field. The study is giving some new information but not contributing to the field significantly. Still, this manuscript should be considered to be very impactful in the field and take the research activities forward in the right direction. The authors are expected to expand the discussion with more details about the increased creatinine as one of the predictive markers. Since CKD is not well associated with PDAC prediction in the samples, it is interesting to know whether there was any AKI was present in these patients. In addition, it is highly imperative that diabetes might associate with any pancreatic disease it might be very difficult to consider this as one of the parameters for differentiating from another pancreatic disease. In addition, the discussion is lacking in correlating the clinical parameters and the disparities between men and women. This is a well-presented manuscript and requires modifications before the final acceptance.
Round 2
Reviewer 2 Report
Authors addressed all the concerns raised by reviewer.